# SEMANTIC MODELING FOR WORLD-CENTERED ARCHITECTURES

**Andrei Mantsivoda & Daria Gavrilina** *
Institute of Mathematics and Information Technologies
Irkutsk State University
Irkutsk, 664003, Russian Federation
`andrei@baikal.ru`

## ABSTRACT

We introduce world-centered multi-agent systems (WMAS) as an alternative to traditional agent-centered architectures, arguing that structured domains such as enterprises and institutional systems require a shared, explicit world representation to ensure semantic consistency, explainability, and long-term stability. We classify worlds along dimensions including ontological explicitness, normativity, etc. In WMAS, learning and coordination operate over a shared world model rather than isolated agent-local representations, enabling global consistency and verifiable system behavior. We propose semantic models as a mathematical formalism for representing such worlds. Finally, we present the Ontobox platform as a realization of WMAS.

## 1 INTRODUCTION

Multi-agent systems (MAS) based on AI are traditionally designed in an agent-centered manner, where each agent maintains its own internal representation of the environment and performs reasoning and learning locally. This paradigm has proven effective in perceptual, exploratory, and adversarial domains, where agents must operate under partial observability, uncertainty, and limited prior structure. However, in domains that require explicit state control, semantic consistency, explainability, and long-term stability – such as organizational, institutional, and enterprise environments – the agent-centered approach reveals significant limitations. In such settings, local world models tend to diverge, learning becomes difficult to verify, and the overall system behavior becomes increasingly opaque as complexity grows.

In our presentation, we introduce and develop the concept of **world-centered multi-agent systems** (WMAS), an architectural paradigm in which the world – rather than individual agents – is treated as the primary representational, learning, and coordination substrate. In WMAS, agents interact with a shared, explicit representation of the world. This shift in perspective enables global semantic consistency, explicit causal reasoning, and verifiable system behavior, addressing fundamental challenges faced by agent-centered MAS in structured domains.

We begin by formalizing what is meant by a "world" in a MAS and propose a classification of worlds based on mathematically relevant properties, including **ontological explicitness, normativity, observability, structural stability**, and **bounded semantic growth**. Based on these dimensions, we distinguish several classes, including perceptual open worlds, hybrid cyber-physical worlds, and **institutional (normative) worlds**. We argue that world-centered architectures are optimal for institutional, normative, semantically unambitious and practically crucial worlds like enterprises, financial systems and healthcare systems.

Within this setting, we define the key conceptual principles of WMAS and contrast them with agent-centered architectures. In WMAS, learning and reasoning are performed with respect to a shared and explicit world representation, while agents – human or artificial – interact with the world through controlled observation and action interfaces. This shift enables shared semantics, global consistency, and explainable behavior, which are difficult to guarantee in agent-centered systems.

---

*

We then propose **semantic models** as a formalism for representing worlds suitable for WMAS. A semantic model consists of two tightly coupled layers. The ground semantic layer is based on object ontologies and represents factual and operational state, including entities, relations, states, and transactions. The causal knowledge layer is based on the logic-probabilistic inference and consists of explicit probabilistic causal relations defined over the ground layer. These relations capture general regularities in the world and are intended to support prediction, diagnosis, and decision-making. **Semantic Machine Learning** (SML) operates over this representation by incrementally updating the causal knowledge layer in response to changes in the ground semantic state. Learning is thus treated as a world-level process rather than an agent-local one. This enables continual adaptation in non-stationary environments while preserving semantic transparency and verifiability.

We argue that semantic models provide a strong rationale for real-world AI deployment in structured domains. They serve simultaneously as operational state representations, causal knowledge bases, and shared context for heterogeneous agents, including symbolic agents and large language models (LLMs). By grounding agent interaction in an explicit world model, semantic models mitigate common issues such as inconsistent beliefs, untraceable decision logic, and unverifiable reasoning. In particular, LLM-based agents can operate as bounded reasoners whose inputs, actions, and explanations are constrained and grounded by the shared semantic world.

Finally, we briefly describe the **Ontobox** (formerly bSystem) platform as a constructive realization of the proposed approach. Ontobox implements semantic models and semantic machine learning as a unified platform for developing world-centered MAS in institutional domains. While implementation details are beyond the scope of this paper, Ontobox demonstrates that the theoretical principles of WMAS can be instantiated in real-world systems, providing an existence proof for the feasibility and practical relevance of the proposed approach.

## 2 WORLD-CENTERED ARCHITECTURES FOR MULTI-AGENT SYSTEMS

### 2.1 FROM AGENT-CENTERED TO WORLD-CENTERED THINKING

Multi-agent systems (MAS) are traditionally designed in an *agent-centered* manner. In such architectures, each agent maintains an internal representation of its environment, performs reasoning locally, and updates its beliefs and policies based on private or partially shared observations. The environment, while formally present, is typically treated as a passive substrate or as a partially observable process whose structure is not globally represented.

This paradigm is effective in perceptual and exploratory domains, where the environment is complex, partially unknown, and must be inferred through interaction. However, in structured domains requiring explicit state control, semantic consistency, and long-term stability—such as enterprises, regulatory systems, or institutional processes—agent-centered architectures exhibit systematic limitations. Independent agent models may diverge, shared semantics may fragment, and causal knowledge becomes difficult to verify or coordinate.

We propose an alternative architectural principle: *world-centered architecture*. In this view, the world is not a passive background but the primary representational and learning substrate. Agents are epistemically and operationally subordinate to a shared, explicit world model. Learning, reasoning, and coordination are defined with respect to this world representation rather than being confined to private agent models.

The feasibility of such an architecture depends entirely on the structural properties of the world. Thus, world-centered design is not universal; it is appropriate only for specific classes of worlds.

### 2.2 WHAT IS A "WORLD" IN A MULTI-AGENT SYSTEM?

In the context of MAS, the world is not the physical environment. It is the totality of entities, relations, states, and norms that agents must reason about, act upon, and coordinate within.

Informally, a world specifies: (i) what exists, (ii) what can change, (iii) what actions are possible, (iv) what constraints apply, (v) what knowledge is shared versus private. More formally, we define a world as follows.

**Definition 1 (World)** *A world $W$ is a tuple*

$$W = (E, R, S, A, T, C),$$

*where:*

- $E$ *is a set of entities,*

- $R$ *is a set of relations over $E$,*

- $S$ *is a state space defined over $(E, R)$,*

- $A$ *is a set of admissible actions or interventions,*

- $T : S \times A \rightarrow S$ *is a transition function or relation,*

- $C$ *is a set of constraints or norms restricting admissible states and transitions.*

A world-centered architecture assumes that $W$ can be represented explicitly and shared among agents. Consequently, the applicability of world-centered MAS depends on whether such an explicit and stable representation is possible.

## 2.3 FUNDAMENTAL DIMENSIONS OF WORLDS

To determine when world-centered architectures are appropriate, we introduce several orthogonal dimensions that capture the structural properties of worlds relevant to architectural design.

**Dimension A: Ontological Explicitness.**  This dimension measures how well-defined "what exists" is in the world.

- **Explicit worlds:** entities and relations are clearly defined and enumerable (e.g., enterprises, legal systems, games).
- **Implicit or emergent worlds:** entities must be inferred from perception (e.g., natural scenes, social discourse).

Ontological explicitness is the most critical condition for world-centered architectures.

**Dimension B: Structural Stability.**  This dimension captures how stable the conceptual schema of the world is over time.

- **Structurally stable worlds:** core concepts evolve slowly and deliberately (institutions, regulatory systems).
- **Structurally fluid worlds:** objects and relations appear or disappear unpredictably (exploration scenarios).

World-centered approaches rely on structural inertia.

**Dimension C: Normativity (Rule-Governedness).**

- **Normative worlds:** actions are governed by explicit rules or constraints (enterprises, finance, games).
- **Non-normative worlds:** dynamics are primarily governed by physics or chance.

Normativity enables symbolic action models and explicit coordination mechanisms.

**Dimension D: Observability and State Accessibility.**

- **State-accessible worlds:** state is inspectable and shareable (databases, digital twins).
- **Partially observable worlds:** state must be inferred indirectly (physical environments).

World-centered architectures require shared state visibility.

**Dimension E: Semantic Ambition (Open-Endedness).**

- **Unambitious worlds:** evolution occurs within known conceptual bounds (accounting systems, supply chains).
- **Ambitious worlds:** fundamentally new object types and relations emerge unpredictably.

World-centered MAS assume bounded semantic growth.

**Dimension F: Perception-to-Deliberation Ratio.**

- **Deliberation-dominant worlds:** complexity lies in reasoning and coordination.
- **Perception-dominant worlds:** complexity lies in sensing and interpretation.

World-centered approaches are most effective in deliberation-dominant worlds.

## 2.4 A CLASSIFICATION OF WORLD TYPES

Using these dimensions, we identify archetypal world classes.

**Type I: Institutional / Enterprise Worlds (Ideal Case).** These worlds exhibit explicit ontology, high structural stability, strong normativity, full state accessibility, bounded semantic evolution, and deliberation dominance. Examples include enterprises, governments, financial systems, and healthcare organizations. World-centered architectures are canonical in this class.

**Type II: Formal Synthetic Worlds (Highly Suitable).** Examples include games and simulated economies. These worlds are explicitly defined and norm-governed, though typically simpler than institutional worlds.

**Type III: Hybrid Cyber-Physical Worlds (Partially Suitable).** Examples include smart factories and traffic systems. These worlds combine explicit symbolic layers with perception-driven subsystems. World-centered models may serve as upper coordination layers.

**Type IV: Perceptual Physical Worlds (Poorly Suited).** Examples include autonomous driving in open environments and robotics in unknown spaces. Ontology is implicit, structure is fluid, and perception dominates. World-centered architectures are not appropriate as primary architectures.

**Type V: Empty or Emerging Worlds (Not Suitable).** Examples include exploratory or creative domains with no predefined ontology. World-centered architectures cannot be instantiated until stable structure emerges.

## 2.5 WHEN IS WORLD-CENTERED ARCHITECTURE APPROPRIATE?

We can now state the applicability condition precisely.

**Definition 2 (Applicability of World-Centered MAS)** *A world-centered architecture is appropriate for a world $W$ when:*

1. *$W$ admits an explicit or explicitly constructible ontology,*

2. *the ontology exhibits structural stability,*

3. *actions are governed by explicit norms or protocols,*

4. *world state is representable and shareable,*

5. *semantic growth is bounded,*

6. *deliberation dominates perception.*

Conversely, world-centered MAS fails when ontology is unknown or continuously mutating, state is fundamentally hidden, or intelligence lies primarily in perception rather than coordination.

## 2.6 Scope and Conceptual Position

World-centered architectures are not universal MAS architectures. They are optimal architectures for a specific but extremely important class of worlds: institutional, normative, and semantically bounded worlds. This reframes world-centered design not as an alternative to agent-centered MAS in general, but as the correct architectural choice for structured, rule-governed domains. Within such worlds, semantic models naturally emerge as suitable representations, as they assume explicit ontology, exploit normativity, manage bounded semantic evolution, and centralize shared state and knowledge. Semantic models are therefore not a general AI solution. They are a world technology for institutional worlds.

## 3 Semantic Models for the World-Centered Architecture

Semantic models constitute a natural formalism for world representation in world-centered architectures because they explicitly encode the structural elements that define a world: entities, relations, states, admissible actions, and normative constraints. In contrast to implicit or agent-local environment models, a semantic model provides a globally shared, formally defined ontology that determines what exists and how it may evolve. This explicit ontological layer ensures semantic consistency across agents and enables the world to function as a single source of truth. Moreover, semantic models support a clear separation between factual state representation and higher-level knowledge about regularities in the world, allowing the architecture to distinguish operational reality from learned abstractions. Such structural transparency is essential in institutional and normative domains, where coordination, compliance, and accountability depend on explicit and verifiable world descriptions.

Beyond static representation, semantic models are particularly well suited to world-centered architectures because they naturally integrate learning at the level of the world itself. By coupling a ground semantic layer with a layer of explicit probabilistic causal relations, semantic models allow world dynamics and causal regularities to be represented within a unified formal framework. Learning mechanisms such as semantic machine learning can then operate directly over the shared world state, updating causal knowledge incrementally as the world evolves. This design aligns with the core principle of world-centered systems: learning modifies the world's knowledge structure rather than isolated agent beliefs. As a result, semantic models provide not only a representation of the world but also a mathematically coherent substrate for explainable, verifiable, and continually adaptive multi-agent coordination in structured environments.

## 3.1 Architecture Overview

The semantic model architecture represents a *two-layer semantic environment* that separates operational state from general, uncertain knowledge, operating within a unified semantic vocabulary.

The first layer represents strict facts, operational parameters, and world structures, while the second layer contains causal knowledge learned from the contents of the first layer. These two layers are tightly coupled through a continual learning mechanism of *semantic machine learning (SML)*.

Figure 1 provides a view of the semantic model architecture and its main components.

In particular, in case of enterprise management systems as examples of institutional worlds, the semantic environment provides a unified knowledge space across all levels of the management system: from the enterprise's transactional activites to agent context support and causal knowledge learning. Specifically, it replaces isolated models of agent's local worlds with a shared, verifiable environment.

## 3.2 Object Ontologies (Ground Semantic Environment)

The first layer of the architecture is the *ground semantic environment*, which represents the operational state of the managed domain and contains basic facts, parameters and structures of the current domain state, including entities, attributes, relations, and available actions. In our architecture this layer is represented by a special type of ontology called *object ontologies*.

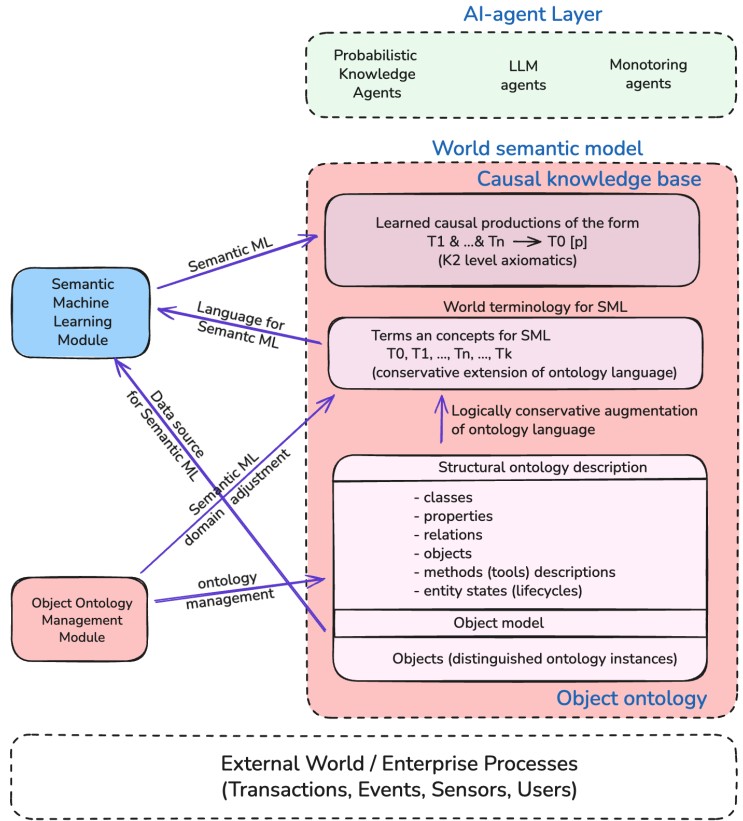

Figure 1: A Semantic Model for the World-Centered Architecture

This layer simultaneously performs several functions: (i) it maintains "factual knowledge" about the current state of the system, (ii) it enforces domain constraints and transactional correctness, (iii) it provides a shared belief context through which agents perceive the environment, (iv) it provides input data for semantic machine learning.

State transitions are executed only by methods (tools), explicitly declared in the object ontology. This ensures that all agent actions remain aligned with world semantics. Thus, the ontology corresponds to a structured, persistent model rather than a passive data store. It provides a unified and coherent space that integrates the knowledge context and tools that operate within it.

### 3.3 LEARNED CAUSAL KNOWLEDGE LAYER

The second layer of the architecture contains a *learned causal knowledge system* defined on top of the ground semantic environment of object ontologies. This layer captures probabilistic regularities, dependencies, and invariants observed in the evolving operational state.

Causal knowledge is represented as a set of logical–probabilistic relations expressed in the same vocabulary as the ground knowledge. Each relation should exceed a reliability threshold, which is a configurable parameter of semantic machine learning. Unlike strict rules, these relations are adaptive and reflect the current operational context. Causal knowledge enables the use of predictive, diagnostic, and decision-support tools.

Importantly, causal knowledge is global and shared across agents. It ensures a consistent interpretation of domain dynamics and supports reasoning under uncertainty.

## 3.4 Semantic Machine Learning (SML)

*SML* is the mechanism that connects the two layers of the semantic environment. SML operates at the object ontology and infers causal relations from the current semantic state.

Learning is continuous and incremental. Changes in the ground semantic environment — such as operational events or state transitions — trigger local updates of the causal knowledge layer. This ensures that the learned knowledge remains synchronized with operational reality without the need for global retraining or system downtime.

By creating verifiable causal relations, SML enables explainability, validation, and human-in-the-loop control, and ensures continual adaptation in non-stationary environments.

## 3.5 Semantic Perception–Action–Learning Loop

During execution, the system evolves through a closed semantic loop: (i) transactions modify the object ontology, (ii) semantic machine learning aligns the causal knowledge layer with the changes, (iii) agents perceive the shared semantic state and causal context, (iv) agents select actions from the tools declared in the ontology, and (v) actions trigger further transactions.

This loop integrates execution, learning, and assessment within a single semantic framework. The ground and learned knowledge is exposed and shared among the agents to make multi-agent behavior more consistent, reduce redundancy in agent reasoning, and provide a stable foundation for long-running enterprise management systems.

## 3.6 Mathematical Background

The general concept of semantic models was introduced in (1). The theory of object ontologies was developed in (4; 6; 7). Semantic machine learning of probabilistic causal relations was developed in (9; 10). The integration of object ontologies and casual machine learning as hybrid semantic models was considered in (8; 5). An efficient algorithm for semantic machine learning was proposed in (2). An approach to access control to semantic structures was developed in (3).

Building on the semantic-model architecture, the next section discusses the interaction of heterogeneous agents, including LLM-based agents, with the semantic environment.

## 4 Multi-Agent and LLM Integration

The architecture described above is designed to support coordination among heterogeneous agents best operating within institutional worlds. In this section, we consider how this architecture plays the role of a shared context within multi-agent interaction and how LLM-based agents are integrated.

The semantic-model architecture provides a strong foundation for building management systems, regardless of whether the agents within it are humans or virtual autonomous actors. However, in the case of AI agents, it is necessary to ensure that they can perceive the environment of a given enterprise, its structure, organization, and operational activities. Thus, figuratively speaking, the semantic environment begins to play a role similar to that of the three-dimensional omnidirectional space in which robots and self-driving cars operate. Instead of a 3D space, our specific space is formed based on the rules and standards in effect at the enterprise. This is a model of the world for agents, uniting all the specific characteristics inherent to the enterprise. Fortunately, this world has important properties: (i) it can be formally covered by regulations and standards and (ii) it is much simpler and more predictable in the context of the operational activities.

## 4.1 Semantic Model as a Coordination Substrate

We propose a semantic model as a tool for constructing a unified space for all agents – both human and autonomous systems – in which each agent's world model is a submodel of the enterprise's shared semantic space. There are three mechanisms of interaction between agents and the environment: (i) data exchange with the object ontology, (ii) obtaining probabilistic knowledge from the causal knowledge layer, and (iii) applying available tools to perform transactions in the ontology.

## 4.2 Heterogeneous Agent Support

The architecture integrates all levels of the world. It provides agents with semantically verified data and accessible tools without intermediaries, reduces inconsistencies between agent beliefs, and simplifies the verification and explanation of agent behavior.

Such a semantic model is designed to support operations at all levels. In particular, in enterprise, it can serve as the knowledge context for heterogeneous agents, including symbolic planners, rule-based agents, optimization-based agents, and learning-based agents. All agents communicate in a common language defined by the object ontology. They can be controlled by uniform behavioral rules determined in the shared environment.

This common language and shared knowledge provide the foundation for collaboration among different agents. Since operational transactions are also formulated in the language of the object ontology, they are also understandable by agents.

## 4.3 Integration of LLM-Based Agents

In such a semantic model, agents typically interact not with the whole model, but with its submodels through a controlled interface. The ontology language allows for strict definition of model access levels using the role mechanism that was developed in (3).

The environment provides LLM-based agents with: (i) snapshots of the current domain state, (ii) access to authorized methods declared in the ontology as tools, and (iii) explicit causal knowledge that can be represented in natural language for reasoning and explanation.

In turn, LLMs provide support for natural language understanding, abstraction, hypothesis generation, and interaction with human users. This preserves the flexibility of language-based intelligence. Grounding LLM operations in the shared object ontology mitigates problems such as hallucination, loss of context, and unverifiable reasoning.

A semantic model also supports explainable interactions between AI agents and human stakeholders. As said before, both factual ontological knowledge and probabilistic knowledge can be transformed into natural language descriptions. This also enables communication between the model and LLMs.

Explanations in a natural language translated from the causal layer are useful for supporting human-in-the-loop control and as common domain knowledge (invariants) for LLMs. Causal relations can thus enhance accountability, verifiability, and trust.

Specifically, if the semantic model is well designed, the generation of an MCP (Model-Context-Protocol) server based on its contextual knowledge and tools can be performed automatically. This makes semantic models a robust technology for the rapid development semantically grounded MCP implementations for various domains. It is also worth noting that semantic models enable the automated generation of APIs alongside OpenAPI descriptions.

Overall, a semantic environment provides a unified interaction framework in which humans and autonomous agents collaborate based on a shared, explainable representation of the domain.

## 5 Implementation and Validation

The semantic model management is implemented in the Ontobox platform. We used Ontobox to evaluate the feasibility of using semantic models as the core technology for large-scale enterprise multi-agent management systems. Ontobox implements the architecture described in sections 3-4 with object ontologies as the operational semantic environment and a continually learned system of probabilistic causal relations.

Ontobox is a platform for developing systems based on the world-centered architecture like enterprise management systems and other norm-governed systems. Overall, the implementation of Ontobox has confirmed the practical viability of the proposed semantic-model architecture for large-scale, real-world multi-agent applications, including a private clinic management system, a car loan management system, an automatic control system for hydroponic farms, etc.

# 6 DISCUSSION: LIMITATIONS AND SCOPE

The proposed architecture was developed to address the specific needs of enterprise multi-agent management systems and other systems with similar properties. Like any architectural approach, it is based on a number of tradeoffs that limit its scope of application. In this section, we discuss key limitations and clarify the intended scope of semantic models.

## 6.1 DOMAIN SCOPE AND APPLICABILITY

We believe that the semantic-model architecture is best suited for *normative, institutionally structured domains*, such as enterprise and organizational management systems – that is, for domains with explicit, stable, and intentionally bounded semantics. These include business process management, resource lifecycle control, financial operations, and compliance-based decision support. The approach demonstrates promising results for highly complex semantic models – the 'glass-box' nature of object ontologies allows the developer to retain full control over such models.

In contrast, domains with poorly defined semantics, constantly changing conceptual structures, or predominantly perceptual inputs may not benefit from the proposed approach. In these cases, the chances of creating and maintaining a coherent semantic environment are significantly lower.

## 6.2 SCALABILITY OF CAUSAL LEARNING

In principle, building probabilistic causal inferences in large semantic environments can lead to combinatorial growth. The proposed model mitigates this risk through a specific design. Semantic machine learning is performed based on a *selected terminology*, which focuses SML only on specific tasks of interest, thus reducing the search space.

Furthermore, learning is incremental and locality-aware. Changes in the operational environment trigger updates only for those causal relations whose semantic reliability is affected, without requiring global retraining. This ensures manageability in large, evolving enterprise environments. Prioritizing carefully selected knowledge over exhaustive causal search is the key idea here.

## 6.3 ADAPTIVITY AND REACTIVITY

SML gradually updates causal knowledge in response to operational changes. This behavior ensures continual adaptation in non-stationary domains, but does not provide instantaneous reactivity.

As a result, the architecture is more suited to analytical and management tasks than to high-frequency control or perception-based tasks.

## 6.4 CONCEPTUAL COMPLEXITY AND ADOPTION

A two-layer semantic architecture appears conceptually more complex than end-to-end learning pipelines. However, in practice, this complexity is primarily concentrated at the initial development stage. In other words, it is limited to system design rather than day-to-day use. Object ontologies closely mirror object-oriented programming models that are familiar to developers and experts. This enables intuitive interaction and visualization (model development tools are a key component of Ontobox). On the other hand, the causal knowledge layer consists of semantically interpretable relations that can be represented in natural language for communication with humans and AI agents.

The same holds true for SML. The main effort is spent at the initial stage on creating a task-specific terminology that determines the learning scope. This step requires significant domain expertise, but reduces ongoing model tuning and debugging. Consequently, the approach trades hidden complexity for explicit and localized knowledge engineering, which is often preferable in enterprise systems.

Developing a specific system on the Ontobox platform resembles traditional system engineering. Specifically, the semantic model can be viewed as an executable technical specification of the system. Undoubtedly, such development requires significant domain expertise – just like creating a traditional system specification. The difference is that once this specification is created, it automatically leads to the creation of the system itself. Thus, complexity is localized in the realm of

knowledge engineering, which is inevitable in case of complex management systems. It also significantly increases developer productivity.

## 6.5 SUMMARY

Overall, the limitations discussed above reflect architectural trade-offs rather than theoretical weaknesses. Semantic models prioritize transparency, shared context, and explainable coordination over universal applicability. They are primarily targeted at applications with explicit, extensive and well-structured knowledge and multi-agent mechanisms.

## 7 CONCLUSION

In this paper we present semantic models as the kernel for the world-centered architecture. This hybrid architecture integrates the system's operational state represented as an object ontology, a causal knowledge layer, and continual semantic machine learning. By separating the ground knowledge of the object ontology and the causal knowledge layer and coupling them through semantic machine learning, we prepare it to serve as manageable and robust shared semantic context for multi-agent systems, providing explainable decision support, and adapting to the non-stationary behavior of management systems.

Future work will focus on further fine-tuning the interaction between the components of the hybrid architecture. The flexibility of semantic models, which are a fusion of efficient object databases and advanced symbolic systems, also encourages us to test a number of hypotheses related to their potential use. Further testing of the architecture is also planned during the development of a number of industrial-grade management systems. From a theoretical perspective, several opportunities exist for the further development of semantic machine learning algorithms and application strategies.

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
