# OpenReview forum: "Semantic Modeling for World-Centered Architectures"
_mathai.club/MathAI/2026/Conference — 2026 Oral_

### Official Review · Reviewer_BKKx · 2026-03-10
**Review on: Semantic Modeling for World-Centered Architectures**

**Rating:** 8
**Confidence:** 5

**Review:**

Paper presents semantic model as the core of a world-aware architecture. Such hybrid architecture integrates the operational state of a system, represented as an object ontology, a causal knowledge layer, and continuous semantic machine learning. By separating the underlying knowledge of the object ontology and the causal knowledge layer and linking them through semantic machine learning, it may be possible to share semantic context for multi-agent systems, providing explainable decision support and adapting to the non-stationary behavior of control systems.

Pros:
- original terminological and mathematical problem statement
- interpretable semantic model as a kernel of the architecture
- potential for efficient multi-agent cooperation within shared domain specific semantic "world model"

Cons:
- no experimental validation presented

---

### Official Review · Reviewer_wyLb · 2026-03-11
**An interesting architectural proposal for world-centered multi-agent systems; well-motivated conceptually, but presented as a position paper without empirical validation or formal theoretical results.**

**Rating:** 4
**Confidence:** 4

**Review:**

## Review

The paper proposes a world-centered multi-agent systems (WMAS) paradigm, in which a shared explicit world model — rather than agent-local representations — serves as the primary substrate for reasoning, learning, and coordination. The framework classifies worlds along six dimensions (ontological explicitness, structural stability, normativity, observability, semantic ambition, perception-to-deliberation ratio) and argues that "institutional worlds" such as enterprises and healthcare systems are well-suited to this approach. A two-layer semantic model (object ontology + causal knowledge) is proposed as the formal representation, with semantic machine learning (SML) as the coupling mechanism. The Ontobox platform is briefly mentioned as a realization.

**Strengths:**

- The central distinction between agent-centered and world-centered architectures is clearly motivated and the identified target class (normative, institutionally structured domains) is sensible and well-scoped.
- The six-dimensional classification of world types is systematic and provides a principled framework for determining architectural applicability.
- The limitations section (Section 6) is honest and well-written, acknowledging scope constraints and architectural trade-offs without overclaiming.

**Weaknesses:**

1. **Absence of empirical evaluation.** Section 5 ("Implementation and Validation") consists of two paragraphs asserting that the Ontobox platform has been applied to a clinic management system, a car loan system, and a hydroponic farm controller. No metrics, no baselines, no comparison with agent-centered alternatives, and no quantitative results are provided. The claim of "confirmed practical viability" is not substantiated. A research paper at this stage requires at minimum a case study with measurable outcomes.

2. **Thin formal content.** The mathematical background (Section 3.6) defers entirely to external references with no formal statements, theorems, or proofs presented in the paper itself. Definition 1 (World as a tuple) is standard and does not constitute a technical contribution. The paper describes an architecture but does not formally analyze it — no correctness properties, no convergence guarantees for SML, no complexity analysis.

3. **Position paper format.** The paper reads as a research vision or system overview rather than a scientific contribution with verifiable results. The structure — motivation, classification, architecture description, platform mention — is characteristic of a workshop position paper or invited system description, not a conference submission requiring original scientific contributions.

4. **Underspecified key components.** The "Semantic Machine Learning" mechanism is central to the architecture but is described only at a high level. How causal relations are induced, what guarantees they satisfy, and how conflicts between the ontology and learned relations are resolved are left unaddressed, with references to external sources that are not summarized.

5. **Relevance to MathAI.** The paper focuses on enterprise and institutional management systems. The connection to mathematical AI, formal reasoning, or explainability in the context of mathematics is not established.

---

## Conclusion

The paper presents a coherent and well-motivated architectural vision for multi-agent systems in normative domains. However, in its current form it does not provide the empirical validation or formal theoretical results expected of a conference publication. It would be better positioned as a position paper, system description, or extended abstract at a workshop. The authors are encouraged to develop the formal foundations and provide a rigorous evaluation before resubmission.

---

### Official Review · Reviewer_n3q9 · 2026-03-12
**Review of Paper: "Semantic Modeling for World-Centered Architectures"**

**Rating:** 6
**Confidence:** 4

**Review:**

This manuscript addresses an important problem in multi-agent systems: how to achieve semantic consistency, explainability, and long-term stability in structured domains such as enterprises and institutions. The authors propose a paradigm shift from agent-centered to world-centered architectures, where agents interact with a shared, explicit world model rather than maintaining private representations. They formalize the notion of a "world" via a tuple (E,R,S,A,T,C) and classify worlds along several dimensions (ontological explicitness, normativity, observability, structural stability, semantic ambition, perception-to-deliberation ratio). Based on this classification, they argue that world-centered architectures are appropriate for institutional worlds with explicit ontologies, stable structures, and strong norms. As a concrete formalism, they introduce semantic models consisting of two layers: (i) a ground semantic layer based on object ontologies (representing factual state and actions) and (ii) a causal knowledge layer capturing probabilistic relations learned from the ground layer via semantic machine learning (SML). The paper also discusses integration with heterogeneous agents, including LLMs, and mentions the Ontobox platform as a proof-of-concept implementation.

### Major Concerns

1. **Mathematical Rigor (Score: 5)**
   The paper provides a formal definition of a world (Definition 1) and an applicability condition for world-centered architectures (Definition 2), which is a positive step. However, these definitions are not used for any subsequent theorems, proofs, or deeper mathematical analysis. The rest of the paper is descriptive and conceptual. There is no formal treatment of learning dynamics, convergence properties of SML, or complexity bounds. The mathematical content remains at an introductory level.

2. **Novelty & Contribution (Score: 6)**
   The idea of a shared world model is not entirely new—it echoes work on semantic web, ontologies, and knowledge graphs. The authors build on a line of previous research (much of it self-cited) spanning decades. The integration with continual learning and LLMs adds some contemporary relevance, but the core concepts (object ontologies, semantic machine learning) appear to be refinements of existing ideas rather than breakthroughs. The paper does not present new algorithms, theorems, or empirical results that would significantly advance the state of the art.

3. **Relevance to MathAI (Score: 8)**
   The topic sits at the intersection of mathematics and AI: formalizing world representations, learning causal structures, and enabling reasoning. The paper addresses foundational issues that are relevant to the conference's scope. However, the lack of mathematical depth limits its impact.

4. **Technical Quality (Score: 5)**
   The methodology is outlined at a conceptual level, but no experiments, benchmarks, or quantitative evaluations are provided. The authors mention several applications (private clinic management, car loan management, hydroponic farms) but give no details about performance, scalability, or comparison with alternative approaches. The description of SML is vague—it is said to be "continuous and incremental" and to use a "reliability threshold," but the underlying algorithm is not explained. The references to previous work (e.g., [2], [5]) suggest that technical details exist elsewhere, but the paper itself does not stand alone.

5. **Clarity & Presentation (Score: 7)**
   The paper is well-structured and clearly written. The definitions and classifications are presented in a logical order. The figures (though not visible in the text) are presumably helpful. Some redundancy and overly general statements could be trimmed. Overall, the exposition is adequate.

6. **AI-Generation Risk (Score: 2)**
   The paper appears to be human-written. It contains specific technical terminology, references to a body of prior work, and a coherent argument that reflects domain expertise. There are no obvious signs of AI-generated content.

### Pros
- Timely and relevant topic for AI in structured domains.
- Clear conceptual framework and classification of worlds.
- Attempt to bridge symbolic AI (ontologies) with learning (causal relations) and modern LLMs.
- Mention of a real-world platform (Ontobox) adds credibility.

### Cons
- Lacks mathematical rigor: no theorems, proofs, or formal analysis.
- Insufficient novelty: builds heavily on prior work without significant new contributions.
- No experimental validation or quantitative results.
- Technical details are missing (e.g., learning algorithm, scalability).
- The paper reads as a position paper or extended abstract rather than a full research article.

### Recommendation
The paper presents an interesting vision but does not meet the standards of a top-tier conference on the mathematics of AI. It would benefit from a more formal treatment of the proposed learning mechanism, theoretical guarantees, and empirical evaluation. In its current form, it is more suitable for a workshop or a vision track.

---

### Official Review · Reviewer_Q1pJ · 2026-03-12
**Semantic Modeling for World-Centered Architectures**

**Rating:** 7
**Confidence:** 4

**Review:**

Summary:
This paper proposes world-centered multi-agent systems (WMAS), an architectural paradigm in which agents interact through a shared semantic representation of the world rather than through independent agent-local models. The world is formally defined as a structured system consisting of entities, relations, states, actions, transition rules, and constraints. The authors introduce semantic models as the formal substrate for representing such worlds and describe a two-layer architecture consisting of object ontologies to represent operational state and a probabilistic causal knowledge layer learned via semantic machine learning. The architecture is implemented in the Ontobox platform and demonstrated in several enterprise-style applications.

Strengths:
- The paper addresses a fundamental issue in multi-agent AI systems: maintaining semantic consistency and explainability in complex multi-agent environments.
- The proposed world-centered perspective provides an interesting alternative to traditional agent-centered architectures.
- The formal definition of worlds and the classification of world types offer a useful conceptual framework for reasoning about multi-agent architectures.
- The separation between operational semantic state and probabilistic causal knowledge supports explainability and verifiable reasoning.
- The Ontobox platform demonstrates the practical feasibility of implementing the proposed architecture in real-world enterprise systems.

Final Recommendation:
Acceptance

---

### Decision · Program_Chairs · 2026-03-14

**Decision:**

Accept (Oral)

**Comment:**

Dear Author(s),

On behalf of the Program Committee of the International Conference on Mathematics of Artificial Intelligence (MathAI 2026), we are pleased to inform you that your paper has been accepted for an oral presentation at MathAI 2026.

Your paper was evaluated through a rigorous two-stage review process involving both automated screening and expert review by members of the Program Committee. The reviewers recognized the quality and contribution of your work.

Presentation details:

- Format: Oral presentation (15–20 minutes + 5 minutes Q&A)
- Mode: You may present either in person (offline) at the conference venue in Sirius, Russia, or remotely via Zoom. Please indicate your preferred mode when confirming your participation.
- Conference dates: Marh 30 - April 3, 2026
- Website: https://mathai.club

Next steps:

1. Please confirm your participation and presentation mode by replying to this email mathai.club@yandex.ru no later than March 15, 2026 18:00 Moscow time.
2. If you plan to attend in person, the organizing committee will provide accommodation details separately.
3. Please prepare your final camera-ready manuscript according to the formatting guidelines available at https://mathai.club and upload it to OpenReview by March 15, 2026 18:00 Moscow time.

Should you have any questions regarding the program, logistics, or your presentation slot, please do not hesitate to contact us.

We look forward to your contribution to MathAI 2026.

With kind regards,

MathAI 2026 Program Committee
International Conference on Mathematics of Artificial Intelligence
https://mathai.club
OpenReview: https://openreview.net/group?id=mathai.club/MathAI/2026/Conference
Telegram: https://t.me/MathAI_club
Email: mathai.club@yandex.ru